# THE CURIOUS CASE OF NEURAL TEXT *De*GENERATION

**Ari Holtzman**[†‡] **Jan Buys**[§†] **Li Du**[†] **Maxwell Forbes**[†‡] **Yejin Choi**[†‡]

[†]Paul G. Allen School of Computer Science & Engineering, University of Washington
[‡]Allen Institute for Artificial Intelligence
[§]Department of Computer Science, University of Cape Town
{ahai,dul2,mbforbes,yejin}@cs.washington.edu, jbuys@cs.uct.ac.za

## ABSTRACT

Despite considerable advances in neural language modeling, it remains an open question what the best *decoding strategy* is for text generation from a language model (e.g. to generate a story). The counter-intuitive empirical observation is that even though the use of likelihood as training objective leads to high quality models for a broad range of language understanding tasks, maximization-based decoding methods such as beam search lead to *degeneration* — output text that is bland, incoherent, or gets stuck in repetitive loops.

To address this we propose **Nucleus Sampling**, a simple but effective method to draw considerably higher quality text out of neural language models than previous decoding strategies. Our approach avoids text *de*generation by truncating the unreliable tail of the probability distribution, sampling from the dynamic nucleus of tokens containing the vast majority of the probability mass.

To properly examine current maximization-based and stochastic decoding methods, we compare generations from each of these methods to the distribution of human text along several axes such as likelihood, diversity, and repetition. Our results show that (1) maximization is an inappropriate decoding objective for open-ended text generation, (2) the probability distributions of the best current language models have an unreliable tail which needs to be truncated during generation and (3) Nucleus Sampling is currently the best available decoding strategy for generating long-form text that is both high-quality — as measured by human evaluation — and as diverse as human-written text.

**Context**: In a shocking finding, scientist discovered a herd of unicorns living in a remote, previously unexplored valley, in the Andes Mountains. Even more surprising to the researchers was the fact that the unicorns spoke perfect English.

**Beam Search, *b*=32**:
"The study, published in the Proceedings of the National Academy of Sciences of the United States of America (PNAS), was conducted by researchers from the Universidad Nacional Autónoma de México (UNAM) and the Universidad Nacional Autónoma de México (UNAM/Universidad Nacional Autónoma de México/Universidad Nacional Autónoma de México/Universidad Nacional Autónoma de México/Universidad Nacional Autónoma de ..."

**Pure Sampling**:
They were cattle called Bolivian Cavalleros; they live in a remote desert uninterrupted by town, and they speak huge, beautiful, paradisiacal Bolivian linguistic thing. They say, 'Lunch, marge.' They don't tell what the lunch is," director Professor Chuperas Omwell told Sky News. "They've only been talking to scientists, like we're being interviewed by TV reporters. We don't even stick around to be interviewed by TV reporters. Maybe that's how they figured out that they're cosplaying as the Bolivian Cavalleros."

Figure 1: Even with substantial human context and the powerful GPT-2 Large language model, Beam Search (size 32) leads to degenerate repetition (highlighted in blue) while pure sampling leads to incoherent gibberish (highlighted in red). When $b \geq 64$, both GPT-2 Large and XL (774M and 1542M parameters, respectively) prefer to stop generating immediately after the given context.

## 1 INTRODUCTION

On February 14th 2019, OpenAI surprised the scientific community with an impressively high-quality article about Ovid's Unicorn, written by GPT-2.[1] Notably, the top-quality generations ob-

---

[1]https://openai.com/blog/better-language-models/

tained from the model rely on *randomness* in the decoding method, in particular through *top-$k$* sampling that samples the next word from the top $k$ most probable choices (Fan et al., 2018; Holtzman et al., 2018; Radford et al., 2019), instead of aiming to decode text that *maximizes* likelihood.

In fact, decoding strategies that optimize for output with high probability, such as beam search, lead to text that is incredibly degenerate, even when using state-of-the-art models such as GPT-2 Large, as shown in Figure 1. This may seem counter-intuitive, as one would expect that good models would assign higher probability to more human-like, grammatical text. Indeed, language models do generally assign high scores to well-formed text, yet the *highest* scores for longer texts are often generic, repetitive, and awkward. Figure 2 exposes how different the distribution of probabilities assigned to beam search decoded text and naturally occurring text really are.

Perhaps equally surprising is the right side of Figure 1, which shows that pure sampling — sampling directly from the probabilities predicted by the model — results in text that is incoherent and almost unrelated to the context. Why is text produced by pure sampling so degenerate? In this work we show that the "unreliable tail" is to blame. This unreliable tail is composed of tens of thousands of candidate tokens with relatively low probability that are over-represented in the aggregate.

To overcome these issues we introduce *Nucleus Sampling* (§3.1). The key intuition of Nucleus Sampling is that the vast majority of probability mass at each time step is concentrated in the *nucleus*, a small subset of the vocabulary that tends to range between one and a thousand candidates. Instead of relying on a fixed top-$k$, or using a temperature parameter to control the shape of the distribution without sufficiently suppressing the unreliable tail, we propose sampling from the top-$p$ portion of the probability mass, expanding and contracting the candidate pool dynamically.

In order to compare current methods to Nucleus Sampling, we compare various distributional properties of generated text to the reference distribution, such as the likelihood of veering into repetition and the perplexity of *generated* text.

The latter reveals that text generated by maximization or top-$k$ sampling is *too* probable, indicating a lack of diversity and divergence in vocabulary usage from the human distribution. On the other hand, pure sampling produces text that is significantly *less* likely than the gold, corresponding to lower generation quality.

Vocabulary usage and Self-BLEU (Zhu et al., 2018) statistics reveal that high values of $k$ are needed to make top-$k$ sampling match human statistics. Yet, generations based on high values of $k$ often have high variance in likelihood, hinting at qualitatively observable incoherency issues. Nucleus Sampling can easily match reference perplexity through tuning the value of $p$, avoiding the incoherence caused by setting $k$ high enough to match distributional statistics.

Finally, we perform Human Unified with Statistical Evaluation (HUSE; Hashimoto et al., 2019) to jointly assess the overall quality and diversity of the decoding strategies, which cannot be captured using either human or automatic evaluation alone. The HUSE evaluation demonstrates that Nucleus Sampling is the best overall decoding strategy. We include generated examples for qualitative analysis – see Figure 3 for a representative example, and further examples in the appendix.[2]

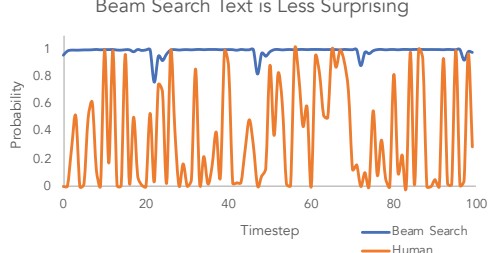

**Beam Search**

...to provide an overview of the current state-of-the-art in the field of computer vision and machine learning, and to provide an overview of the current state-of-the-art in the field of computer vision and machine learning, and to provide an overview of the current state-of-the-art in the field of computer vision and machine learning, and to provide an overview of the current state-of-the-art in the field of computer vision and machine learning, and...

**Human**

...which grant increased life span and three years warranty. The Antec HCG series consists of five models with capacities spanning from 400W to 900W. Here we should note that we have already tested the HCG-620 in a previous review and were quite satisfied With its performance. In today's review we will rigorously test the Antec HCG-520, which as its model number implies, has 520W capacity and contrary to Antec's strong beliefs in multi-rail PSUs is equipped...

Figure 2: The probability assigned to tokens generated by Beam Search and humans, given the same context. Note the increased variance that characterizes human text, in contrast with the endless repetition of text decoded by Beam Search.

---

[2]Code and all generations are available at `https://github.com/ari-holtzman/degen`

## 2 BACKGROUND

### 2.1 TEXT GENERATION DECODING STRATEGIES

A number of recent works have alluded to the disadvantages of generation by maximization, which tend to generate output with high grammaticality but low diversity (Kulikov et al., 2019; Holtzman et al., 2018; Fan et al., 2018). Generative Adversarial Networks (GANs) have been a prominent research direction (Yu et al., 2017; Xu et al., 2018), but recent work has shown that when quality and diversity are considered jointly, GAN-generated text fails to outperform generations from language models (Caccia et al., 2018; Tevet et al., 2019; Semeniuta et al., 2018). Work on neural dialog systems have proposed methods for diverse beam search, using a task-specific diversity scoring function or constraining beam hypotheses to be sufficiently different (Li et al., 2016a; Vijayakumar et al., 2018; Kulikov et al., 2019; Pal et al., 2006). While such utility functions encourage desirable properties in generations, they do not remove the need to choose an appropriate decoding strategy, and we believe that Nucleus Sampling will have complementary advantages in such approaches. Finally, Welleck et al. (2020) begin to address the problem of neural text degeneration through an "unlikelihood loss", which decreases training loss on repeated tokens and thus implicitly reduces gradients on frequent tokens as well. Our focus is on exposing neural text degeneration and providing a *decoding* solution that can be used with arbitrary models, but future work will likely combine training-time and inference-time solutions.

### 2.2 OPEN-ENDED VS DIRECTED GENERATION

Many text generation tasks are defined through (input, output) pairs, such that the output is a constrained *transformation* of the input. Example applications include machine translation (Bahdanau et al., 2015), data-to-text generation (Wiseman et al., 2017), and summarization (Nallapati et al., 2016). We refer to these tasks as *directed* generation. Typically encoder-decoder architectures are used, often with an attention mechanism (Bahdanau et al., 2015; Luong et al., 2015) or using attention-based architectures such as the Transformer (Vaswani et al., 2017). Generation is usually performed using beam search; since output is tightly scoped by the input, repetition and genericness are not as problematic. Still, similar issues have been reported when using large beam sizes (Koehn & Knowles, 2017) and more recently with exact inference (Stahlberg & Byrne, 2019), a counter-intuitive observation since more comprehensive search helps maximize probability.

*Open-ended generation*, which includes conditional story generation and contextual text continuation (as in Figure 1), has recently become a promising research direction due to significant advances in neural language models (Clark et al., 2018; Holtzman et al., 2018; Fan et al., 2018; Peng et al., 2018; Radford et al., 2019). While the input context restricts the space of acceptable output generations, there is a considerable degree of freedom in what can plausibly come next, unlike in directed generation settings. Our work addresses the challenges faced by neural text generation with this increased level of freedom, but we note that some tasks, such as goal-oriented dialog, may fall somewhere in between open-ended and directed generation.

## 3 LANGUAGE MODEL DECODING

Given an input text passage as context, the task of *open-ended* generation is to generate text that forms a coherent continuation from the given context. More formally, given a sequence of $m$ tokens $x_1 \ldots x_m$ as **context**, the task is to generate the next $n$ **continuation** tokens to obtain the completed sequence $x_1 \ldots x_{m+n}$. We assume that models compute $P(x_{1:m+n})$ using the common left-to-right decomposition of the text probability,

$$P(x_{1:m+n}) = \prod_{i=1}^{m+n} P(x_i | x_1 \ldots x_{i-1}), \tag{1}$$

which is used to generate the generation token-by-token using a particular *decoding strategy*.

**Maximization-based decoding** The most commonly used decoding objective, in particular for directed generation, is maximization-based decoding. Assuming that the model assigns higher probability to higher quality text, these decoding strategies search for the continuation with the highest

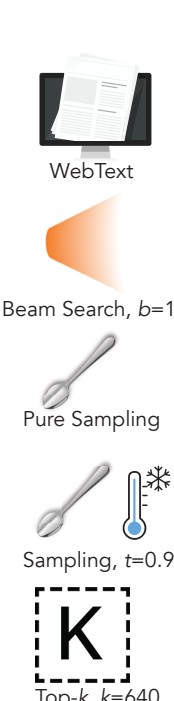

**WebText**

**An unprecedented number of mostly young whales have become stranded on the West Australian coast since 2008.**

**Beam Search, b=16**

The number of stranded whales has increased by more than 50 per cent in the past year, with the number of stranded whales on the West Australian coast increasing by more than 50 per cent in the past year. The number of whales stranded on the West Australian coast has increased by more than 50 per cent in the past year, with the number of stranded whales on the West Australian coast increasing by more than 50 per cent in the past year.

**Pure Sampling**

The Australian Food Safety Authority has warned Australia's beaches may be revitalised this year because healthy seabirds and seals have been on the move. More than 50,000 seabirds, sea mammals and seahorses have been swept into the sea by the Holden CS118 and Adelaide Airport CS300 from 2013. A major white-bat and umidauda migration across Australia is under way in Australia for the first time, with numbers reaching an estimated 50,000.

**Sampling, t=0.9**

Last week's intense storms and a series of powerful cyclones have been officially blamed for the deaths of at least nine large fin whales near Whitsundays - the largest loss of any species globally. The fin whales: packed in the belly of one killer whale thrashing madly in fear as another tries to bring it to safety. When the colossal animal breached the waters of Whitsundays, he'd been seen tagged for a decade.

**Top-k, k=640**

Pumping Station #3 shut down due to construction damage Find more at: www.abc.net.au/environment/species-worry/ in-the-top-10-killer-whale-catastrophes-in-history.html "In the top 10 killer whale catastrophes in history: 1) 1986: Up to 12 orcas struck by lightning; many drowned and many more badly injured.

**Top-k, k=40, t=0.7**

The whale's fate was confirmed late last week when the animal was found by fishermen off the coast of Bundaberg. Experts believe the whale was struck by a fishing vessel off the coast of Bundaberg, and died after being sucked into the ocean. The whale's fate was confirmed late last week when the animal was found by fishermen off the coast of Bundaberg.

**Nucleus, p=0.95**

There has been an unprecedented number of calves caught in the nets of whaling stations that operate in WA. Pilot whales continue to migrate to feeding grounds to feed their calves. They are now vulnerable due to the decline of wild populations; they are restricted to one breeding site each year. Image copyright Yoon Bo Kim But, with sharp decline in wild populations the size of the Petrels are shrinking and dwindling population means there will only be room for a few new fowl.

**WebText**

Poor nutrition has led to a rise in the number of stranded humpback whales on the West Australian coast, veterinary researchers have said. Carly Holyoake, from Murdoch University, at the Australian Veterinary Association's annual conference in Perth on Wednesday, said an unprecedented number of mostly young whales had become stranded on the coast since 2008.

Figure 3: Example generations continuing an initial sentence. Maximization and top-$k$ truncation methods lead to copious repetition (highlighted in blue), while sampling with and without temperature tends to lead to incoherence (highlighted in red). Nucleus Sampling largely avoids both issues.

likelihood. Since finding the optimum argmax sequence from recurrent neural language models or Transformers is not tractable (Chen et al., 2018), common practice is to use beam search (Li et al., 2016b; Shen et al., 2017; Wiseman et al., 2017). However, several recent studies on open-ended generation have reported that maximization-based decoding does not lead to high quality text (Fan et al., 2018; Holtzman et al., 2018).

## 3.1 NUCLEUS SAMPLING

We propose a new stochastic decoding method: Nucleus Sampling. The key idea is to use the shape of the probability distribution to determine the set of tokens to be sampled from. Given a distribution $P(x|x_{1:i-1})$, we define its top-$p$ vocabulary $V^{(p)} \subset V$ as the smallest set such that

$$\sum_{x \in V^{(p)}} P(x|x_{1:i-1}) \geq p. \tag{2}$$

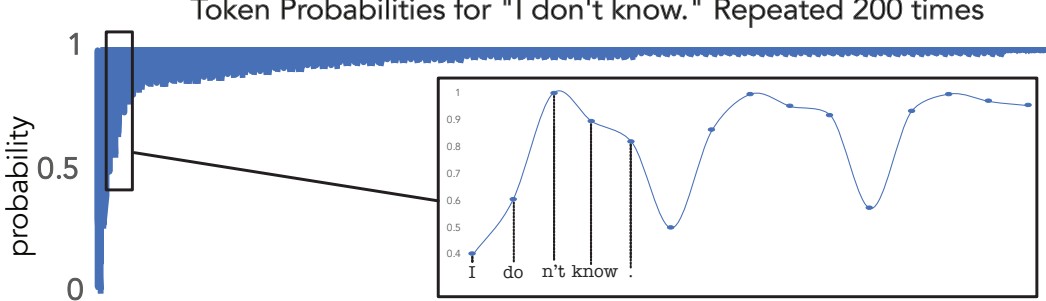

Figure 4: The probability of a repeated phrase increases with each repetition, creating a positive feedback loop. We found this effect to hold for the vast majority of phrases we tested, regardless of phrase length or if the phrases were sampled randomly rather than taken from human text.

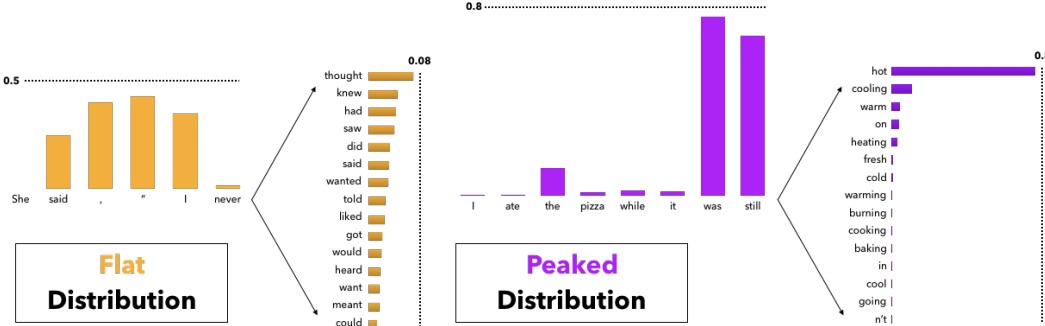

Figure 5: The probability mass assigned to partial human sentences. Flat distributions lead to many moderately probable tokens, while peaked distributions concentrate most probability mass into just a few tokens. The presence of flat distributions makes the use of a small $k$ in top-$k$ sampling problematic, while the presence of peaked distributions makes large $k$'s problematic.

Let $p' = \sum_{x \in V^{(p)}} P(x|x_{1:i-1})$. The original distribution is re-scaled to a new distribution, from which the next word is sampled:

$$P'(x|x_{1:i-1}) = \begin{cases} P(x|x_{1:i-1})/p' & \text{if } x \in V^{(p)} \\ 0 & \text{otherwise.} \end{cases} \tag{3}$$

In practice this means selecting the highest probability tokens whose cumulative probability mass exceeds the pre-chosen threshold $p$. The size of the sampling set will adjust dynamically based on the shape of the probability distribution at each time step. For high values of $p$, this is a small subset of vocabulary that takes up vast majority of the probability mass — the *nucleus*.

### 3.2 TOP-$k$ SAMPLING

Top-$k$ sampling has recently become a popular alternative sampling procedure (Fan et al., 2018; Holtzman et al., 2018; Radford et al., 2019). Nucleus Sampling and top-$k$ both sample from truncated Neural LM distributions, differing only in the strategy of where to truncate. Choosing where to truncate can be interpreted as determining the generative model's trustworthy prediction zone.

At each time step, the top $k$ possible next tokens are sampled from according to their relative probabilities. Formally, given a distribution $P(x|x_{1:i-1})$, we define its top-$k$ vocabulary $V^{(k)} \subset V$ as the set of size $k$ which maximizes $\sum_{x \in V^{(k)}} P(x|x_{1:i-1})$. Let $p' = \sum_{x \in V^{(k)}} P(x|x_{1:i-1})$. The distribution is then re-scaled as in equation 3, and sampling is performed based on that distribution. Note that the scaling factor $p'$ can vary wildly at each time-step, in contrast to Nucleus Sampling.

**Difficulty in choosing a suitable value of $k$**   While top-$k$ sampling leads to considerably higher quality text than either beam search or sampling from the full distribution, the use of a constant $k$ is

sub-optimal across varying contexts. As illustrated on the left of Figure 5, in some contexts the head of the next word distribution can be flat across tens or hundreds of reasonable options (e.g. nouns or verbs in generic contexts), while in other contexts most of the probability mass is concentrated in one or a small number of tokens, as on the right of the figure. Therefore if $k$ is small, in some contexts there is a risk of generating bland or generic text, while if $k$ is large the top-$k$ vocabulary will include inappropriate candidates which will have their probability of being sampled *increased* by the renormalization. Under Nucleus Sampling, the number of candidates considered rises and falls dynamically, corresponding to the changes in the model's confidence region over the vocabulary which top-$k$ sampling fails to capture for any one choice of $k$.

### 3.3 Sampling with Temperature

Another common approach to sampling-based generation is to shape a probability distribution through temperature (Ackley et al., 1985). Temperature sampling has been applied widely to text generation (Ficler & Goldberg, 2017; Fan et al., 2018; Caccia et al., 2018). Given the logits $u_{1:|V|}$ and temperature $t$, the softmax is re-estimated as

$$p(x = V_l | x_{1:i-1}) = \frac{\exp(u_l/t)}{\sum_{l'} \exp(u'_l/t)}. \tag{4}$$

Setting $t \in [0, 1)$ skews the distribution towards high probability events, which implicitly lowers the mass in the tail distribution. Low temperature sampling has also been used to partially alleviate the issues of top-$k$ sampling discussed above, by shaping the distribution before top-$k$ sampling (Radford et al., 2018; Fan et al., 2018). However, recent analysis has shown that, while lowering the temperature improves generation quality, it comes at the cost of decreasing diversity (Caccia et al., 2018; Hashimoto et al., 2019).

## 4 Likelihood Evaluation

### 4.1 Experimental Setup

While many neural network architectures have been proposed for language modeling, including LSTMs (Sundermeyer et al., 2012) and convolutional networks (Dauphin et al., 2017), the Transformer architecture (Vaswani et al., 2017) has been the most successful in the extremely large-scale training setups in recent literature (Radford et al., 2018; 2019). In this study we use the Generatively Pre-trained Transformer, version 2 (GPT2; Radford et al., 2019), which was trained on WebText, a 40GB collection of text scraped from the web.[3] We perform experiments using the Large model (762M parameters). Our analysis is based on generating 5,000 text passages, which end upon reaching an end-of-document token or a maximum length of 200 tokens. Texts are generated conditionally, conditioned on the initial paragraph (restricted to 1-40 tokens) of documents in the held-out portion of WebText, except where otherwise mentioned.

### 4.2 Perplexity

Our first evaluation is to compute the perplexity of *generated* text using various decoding strategies, according to the model that is being generated from. We compare these perplexities against that of the gold text (Figure 6). Importantly, we argue that the optimal generation strategy should produce text which has a perplexity *close to* that of the gold text: Even though the model has the ability to generate text that has lower perplexity (higher probability), such text tends to have low diversity and get stuck in repetition loops, as shown in §5 and illustrated in Figure 4.

We see that perplexity of text obtained from pure sampling is *worse* than the perplexity of the gold. This indicates that the model is confusing itself: sampling too many unlikely tokens and creating context that makes it difficult to recover the human distribution of text, as in Figure 1. Yet, setting the temperature lower creates diversity and repetition issues, as we shall see in §5. Even with our relatively fine-grained parameter sweep, Nucleus Sampling obtains closest perplexity to human text, as shown in Table 1.

---

[3]Available at `https://github.com/openai/gpt-2-output-dataset`

| Method | Perplexity | Self-BLEU4 | Zipf Coefficient | Repetition % | HUSE |
|---|---|---|---|---|---|
| Human | 12.38 | 0.31 | 0.93 | 0.28 | - |
| Greedy | 1.50 | 0.50 | 1.00 | 73.66 | - |
| Beam, b=16 | 1.48 | 0.44 | 0.94 | 28.94 | - |
| Stochastic Beam, b=16 | 19.20 | 0.28 | 0.91 | 0.32 | - |
| Pure Sampling | 22.73 | 0.28 | **0.93** | 0.22 | 0.67 |
| Sampling, $t$=0.9 | 10.25 | 0.35 | 0.96 | 0.66 | 0.79 |
| Top-$k$=40 | 6.88 | 0.39 | 0.96 | 0.78 | 0.19 |
| Top-$k$=640 | 13.82 | **0.32** | 0.96 | **0.28** | 0.94 |
| Top-$k$=40, $t$=0.7 | 3.48 | 0.44 | 1.00 | 8.86 | 0.08 |
| Nucleus $p$=0.95 | **13.13** | **0.32** | 0.95 | 0.36 | **0.97** |

Table 1: Main results for comparing all decoding methods with selected parameters of each method. The numbers *closest to human scores* are in **bold** except for HUSE (Hashimoto et al., 2019), a combined human and statistical evaluation, where the highest (best) value is **bolded**. For Top-$k$ and Nucleus Sampling, HUSE is computed with interpolation rather than truncation (see §6.1).

### 4.3 NATURAL LANGUAGE DOES NOT MAXIMIZE PROBABILITY

One might wonder if the issue with maximization is a *search error*, i.e., there are higher quality sentences to which the model assigns higher probability than to the decoded ones, beam search has just failed to find them. Yet Figures 2 & 6 show that the per-token probability of natural text is, on average, much *lower* than text generated by beam search. Natural language rarely remains in a high probability zone for multiple consecutive time steps, instead veering into lower-probability but more informative tokens. Nor does natural language tend to fall into repetition loops, even though the model tends to assign high probability to this, as seen in Figure 4.

Why is human-written text *not* the most probable text? We conjecture that this is an intrinsic property of human language. Language models that assign probabilities one word at a time without a global model of the text will have trouble capturing this effect. Grice's Maxims of Communication (Grice, 1975) show that people optimize against stating the obvious. Thus, making every word as predictable as possible will be disfavored. This makes solving the problem simply by training larger models or improving neural architectures using standard per-word learning objectives unlikely: such models are forced to favor the lowest common denominator, rather than informative language.

## 5 DISTRIBUTIONAL STATISTICAL EVALUATION

### 5.1 ZIPF DISTRIBUTION ANALYSIS

In order to compare generations to the reference text, we begin by analyzing their use of vocabulary. Zipf's law suggests that there is an exponential relationship between the rank of a word and its frequency in text. The Zipfian coefficient $s$ can be used to compare the distribution in a given text

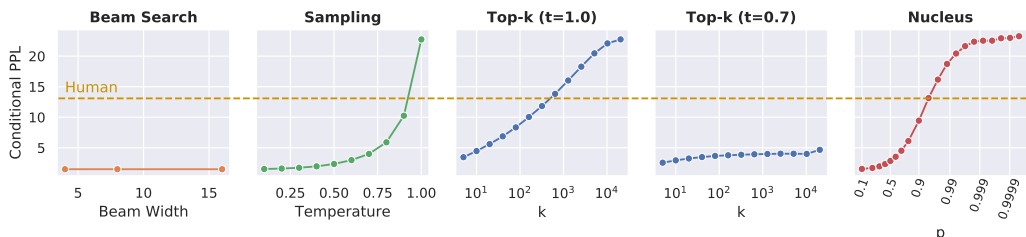

Figure 6: Perplexities of generations from various decoding methods. Note that beam search has unnaturally low perplexities. A similar effect is seen using a temperature of $0.7$ with top-$k$ as in both Radford et al. (2019) and Fan et al. (2018). Sampling, Top-$k$, and Nucleus can all be calibrated to human perplexities, but the first two face coherency issues when their parameters are set this high.

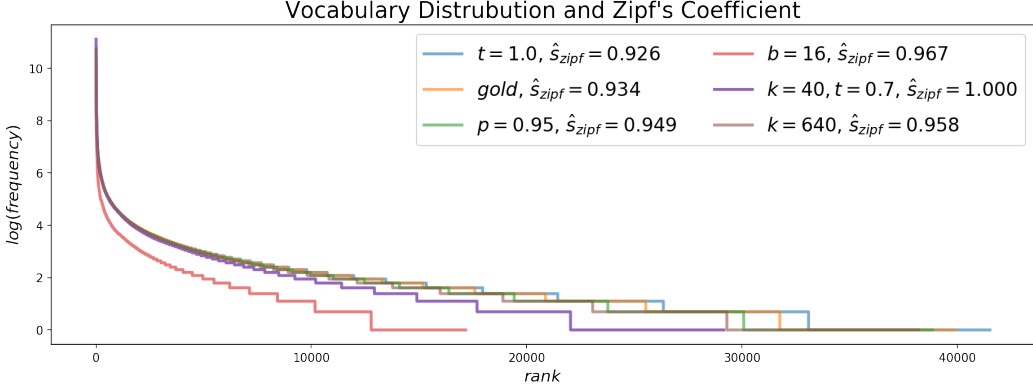

Figure 7: A rank-frequency plot of the distributional differences between $n$-gram frequencies of human and machine text. Sampling and Nucleus Sampling are by far the closest to the human distribution, while Beam Search clearly follows a very different distribution than natural language.

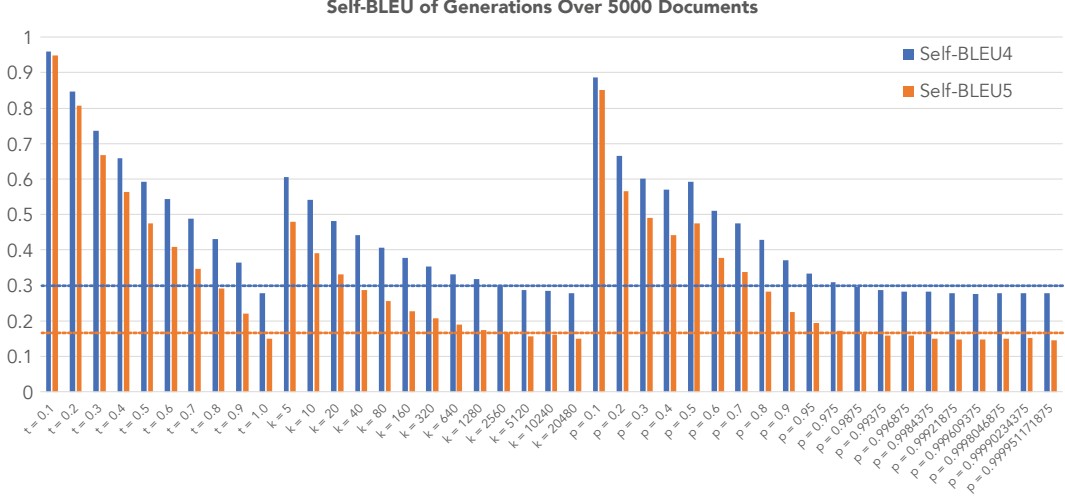

Figure 8: Self-BLEU calculated on the unconditional generations produced by stochastic decoding methods; lower Self-BLEU scores imply higher diversity. Horizontal blue and orange lines represent human self-BLEU scores. Note how common values of $t \in [0.5, 1]$ and $k \in [1, 100]$ result in high self-similarity, whereas "normal" values of $p \in [0.9, 1)$ closely match the human distribution of text.

to a theoretically perfect exponential curve, where $s = 1$ (Piantadosi, 2014). Figure 7 shows the vocabulary distributions along with estimated Zipf coefficients for selected parameters of different decoding methods. As expected, pure sampling is the closest to the human distribution, followed by Nucleus Sampling. The visualization of the distribution shows that pure sampling slightly *overestimates* the use of rare words, likely one reason why pure sampling also has higher perplexity than human text. Furthermore, lower temperature sampling avoids sampling these rare words from the tail, which is why it has been used in some recent work (Fan et al., 2018; Radford et al., 2019).

## 5.2 SELF-BLEU

We follow previous work and compute Self-BLEU (Zhu et al., 2018) as a metric of diversity. Self-BLEU is calculated by computing the BLEU score of each generated document using *all other generations* in the evaluation set as references. Due to the expense of computing such an operation, we sample 1000 generations, each of which is compared with *all 4999 other generations as references*. A lower Self-BLEU score implies higher diversity. Figure 8 shows that Self-BLEU results largely follow that of the Zipfian distribution analysis as a diversity measure. It is worth noting that

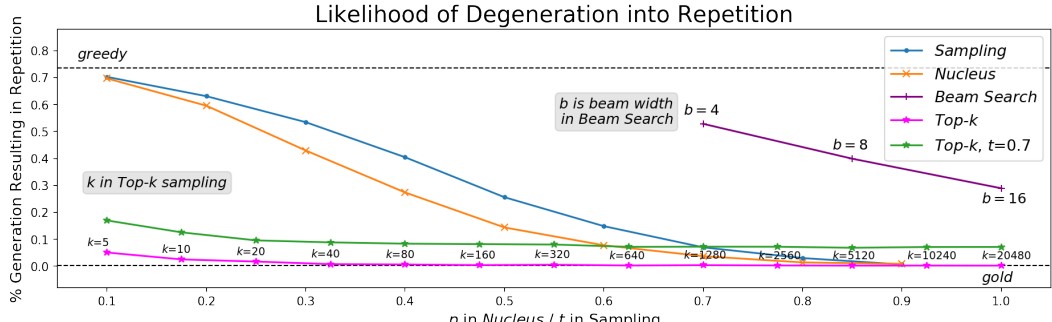

Figure 9: We visualize how often different decoding methods get "stuck" in loops within the first 200 tokens. A phrase (minimum length 2) is considered a repetition when it repeats at least **three** times at the *end* of the generation. We label points with their parameter values except for $t$ and $p$ which follow the x-axis. Values of $k$ greater than 100 are rarely used in practice and values of $p$ are usually in $[0.9, 1)$; therefore Nucleus Sampling is far closer to the human distribution in its usual parameter range. Sampling with temperatures lower than 0.9 severely increase repetition. Finally, although beam search becomes less repetitive according to this metric as beam width increases, this is largely because average length gets shorter as $b$ increases (see Appendix A).

very high values of $k$ and $t$ are needed to get close to the reference distribution, though these result in unnaturally high perplexity (§4).

## 5.3 REPETITION

One attribute of text quality that we can quantify is repetition. Figure 9 shows that Nucleus Sampling and top-$k$ sampling have the least repetition for reasonable parameter ranges. Generations from temperature sampling have more repetition unless very high temperatures are used, which we have shown negatively affects coherence (as measured by high perplexity). Further, all stochastic methods face repetition issues when their tuning parameters are set too low, which tends to *over-truncate*, mimicking greedy search. Therefore we conclude that only Nucleus Sampling satisfies all the distributional criteria for desirable generations.

## 6 HUMAN EVALUATION

### 6.1 HUMAN UNIFIED WITH STATISTICAL EVALUATION (HUSE)

Statistical evaluations are unable to measure the coherence of generated text properly. While the metrics in previous sections gave us vital insights into the different decoding methods we compare, human evaluation is still required to get a full measure of the quality of the generated text. However, pure human evaluation does not take into account the diversity of the generated text; therefore we use HUSE (Hashimoto et al., 2019) to combine human and statistical evaluation. HUSE is computed by training a discriminator to distinguish between text drawn from the human and model distributions, based on only two features: The probability assigned by the language model, and human judgements of typicality of generations. Text that is close to the human distribution in terms of quality and diversity should perform well on both likelihood evaluation and human judgements.

As explored in the previous sections, the current best-performing decoding methods rely on *truncation* of the probability distribution, which yields a probability of 0 for the vast majority of potential tokens. Initial exploration of applying HUSE directly led to top-$k$ and Nucleus Sampling receiving scores of nearly 0 due to truncation, despite humans favoring these methods. As a proxy, when generating the text used to compute HUSE, we interpolate (with mass $0.1$) the original probability distribution with the top-$k$ and Nucleus Sampling distribution, smoothing the truncated distribution.

For each decoding algorithm we annotate 200 generations for typicality, with each generation receiving 20 annotations from 20 different annotators. This results in a total of 4000 annotations per a

decoding scheme. We use a KNN classifier to compute HUSE, as in the original paper, with $k = 13$ neighbors, which we found led to the higher accuracy in discrimination. The results in Table 1 shows that Nucleus Sampling obtains the highest HUSE score, with Top-$k$ sampling performing second best.

## 6.2 QUALITATIVE ANALYSIS

Figure 3 shows representative example generations. Unsurprisingly, beam search gets stuck in a repetition loop it cannot escape. Of the stochastic decoding schemes, the output of full sampling is clearly the hardest to understand, even inventing a new word "umidauda", apparently a species of bird. The generation produced by Nucleus Sampling isn't perfect – the model appears to confuse whales with birds, and begins writing about those instead. Yet, top-$k$ sampling immediately veers off into an unrelated event. When top-$k$ sampling is combined with a temperature of 0.7, as is commonly done (Radford et al., 2019; Fan et al., 2018), the output devolves into repetition, exhibiting the classic issues of low-temperature decoding. More generations are available in Appendix B.

## 7 CONCLUSION

This paper provided a deep analysis into the properties of the most common decoding methods for open-ended language generation. We have shown that likelihood maximizing decoding causes repetition and overly generic language usage, while sampling methods without truncation risk sampling from the low-confidence tail of a model's predicted distribution. Further, we proposed Nucleus Sampling as a solution that captures the region of confidence of language models effectively. In future work, we wish to dynamically characterize this region of confidence and include a more semantic utility function to guide the decoding process.

## ACKNOWLEDGMENTS

This research was supported in part by NSF (IIS-1524371), the National Science Foundation Graduate Research Fellowship under Grant No. DGE1256082, DARPA CwC through ARO (W911NF15-1- 0543), DARPA MCS program through NIWC Pacific (N66001-19-2-4031), the South African Centre for Artificial Intelligence Research, and the Allen Institute for AI.

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

# A  BEAM WIDTH EFFECT

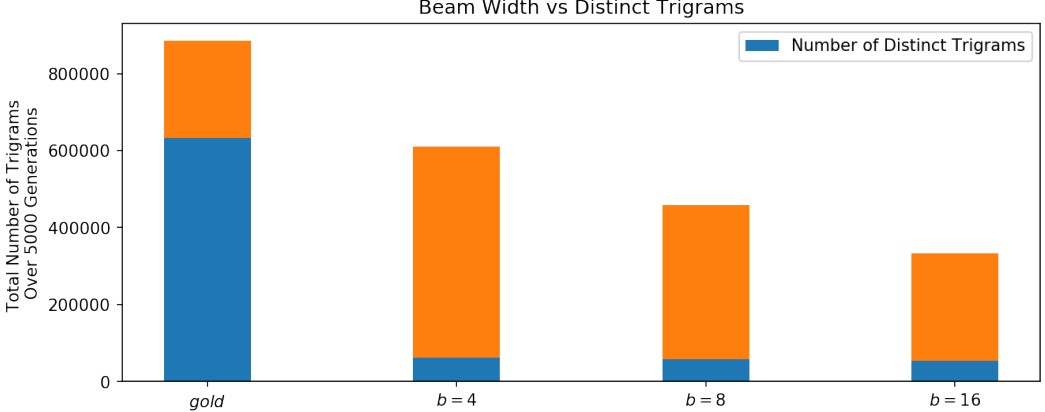

Figure 10: The total number of trigrams produced by Beam Search with varying beam widths, with gold (human) data for comparison. Note how the average length of generations goes down linearly with beam width, while the number of distinct trigrams stays constant and extremely low in comparison to gold data.

## B   EXAMPLE GENERATIONS

We include a set of examples for further qualitative comparison.

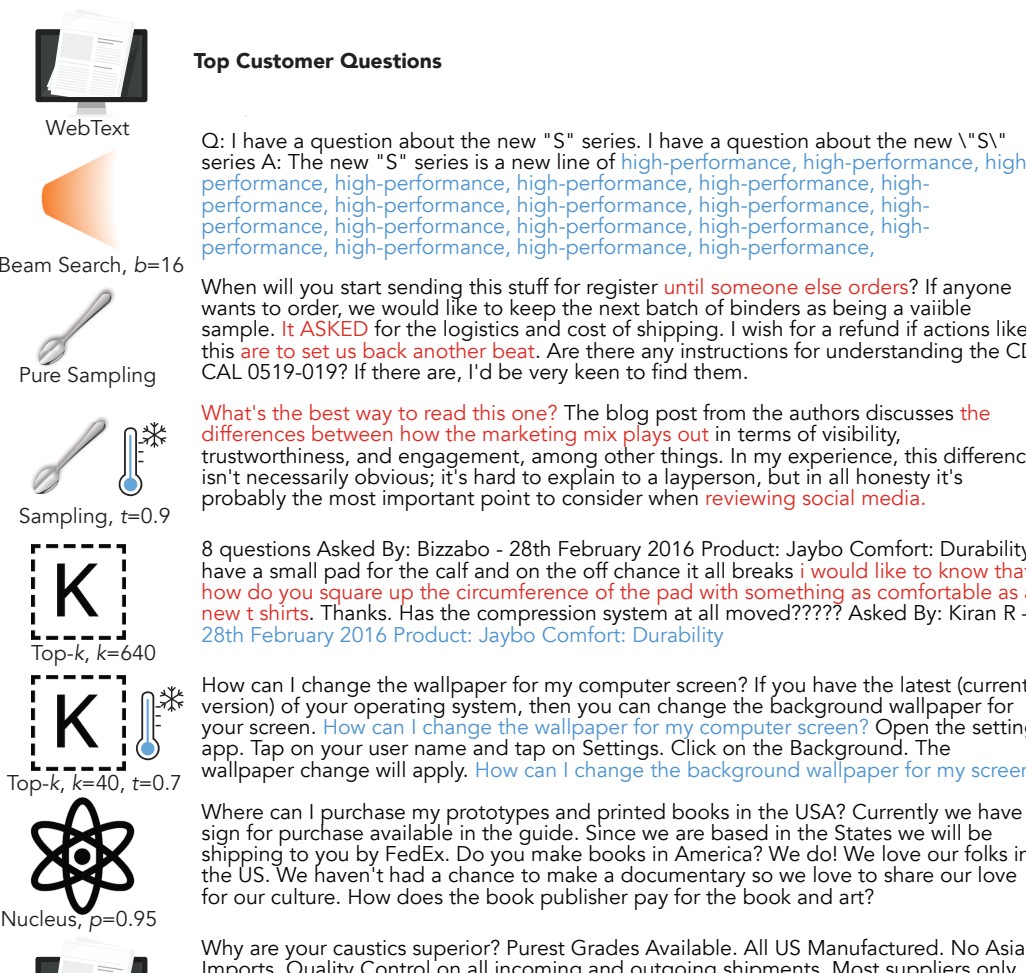

Figure 11: More example generations from an initial tag line. All generations available at `https://github.com/ari-holtzman/degen`

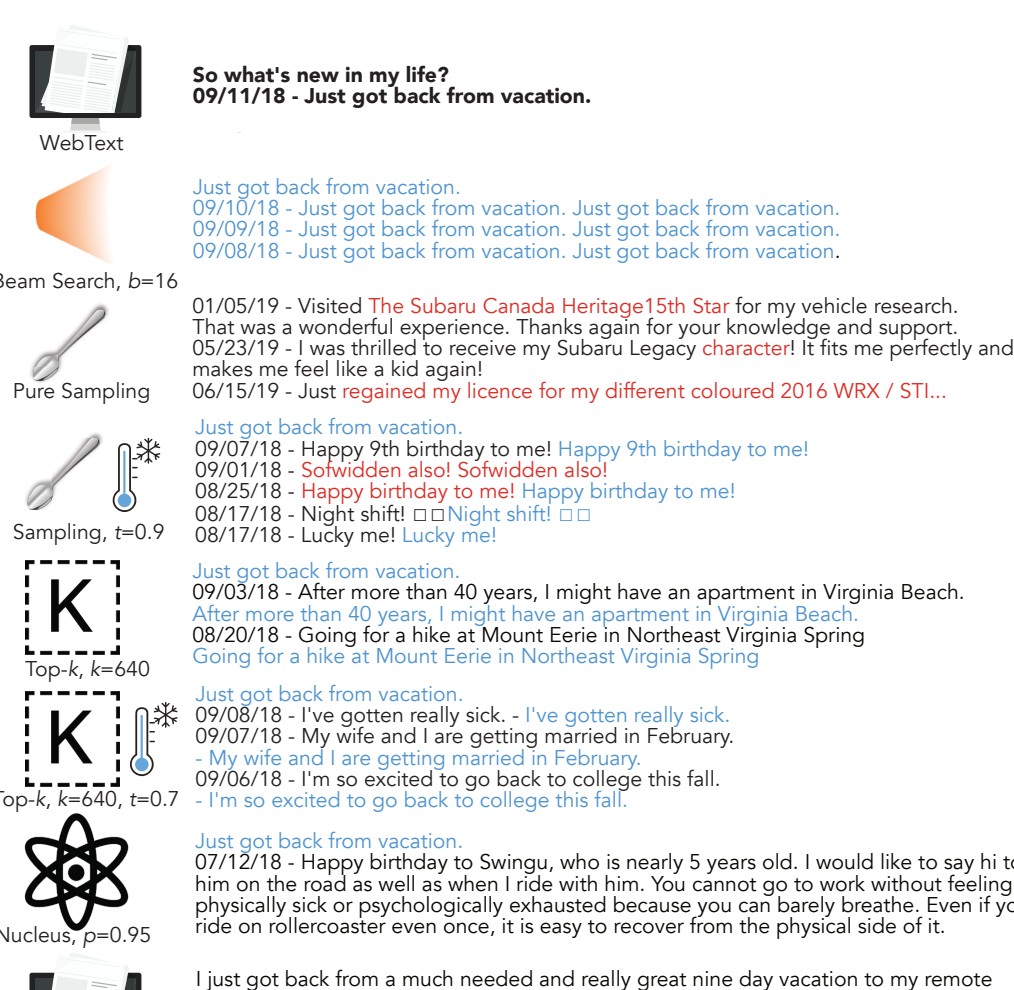

Figure 12: More example generations from an initial tag line. Note that Pure Sampling and Nucleus Sampling is the only algorithms that can escape the repetition loop, with Nucleus Sampling's generation far closer in style to the ground truth text. All generations available at `https://github.com/ari-holtzman/degen`

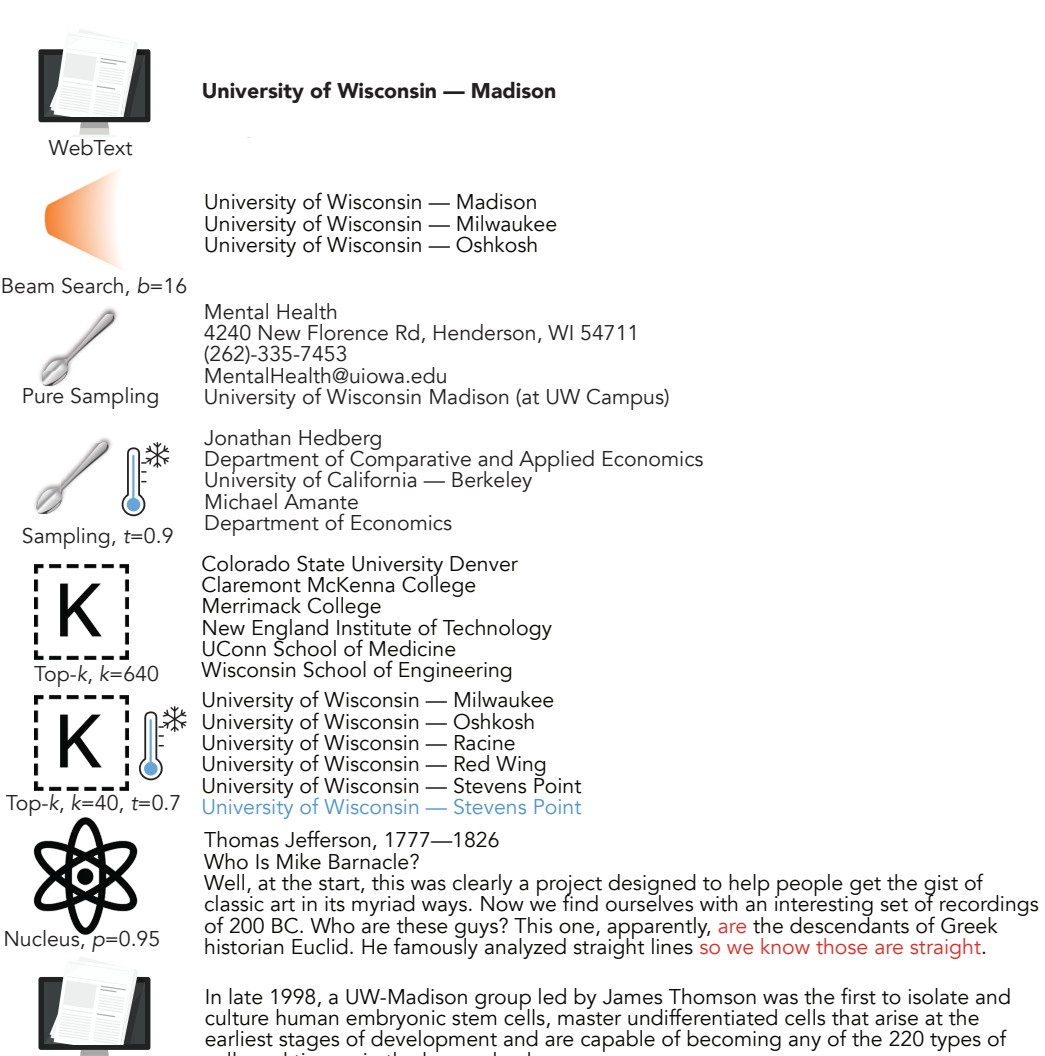

Figure 13: More example generations from an initial tag line. All generations available at `https://github.com/ari-holtzman/degen`

