# OpenReview forum: "The Curious Case of Neural Text Degeneration"
_ICLR.cc/2020/Conference — Accept (Poster)_

### Official Review · AnonReviewer2 · 2019-10-22
**Official Blind Review #2**

**Rating:** 6

**Review:**

Contributions:

This paper studies an important problem, i.e., how to find a good decoding strategy for open-ended text generation. To this end, the authors provide a deep analysis of the most common decoding methods, and propose Nucleus Sampling, a very simple yet effective method to generate higher-quality text. Compared with top-k sampling, the key idea behind the proposed method is to sample from the dynamic nucleus of tokens containing the majority of the probability mass. Experiments demonstrate that nucleus sampling is an effective decoding strategy in practice.

Strengths:

(1) Writing & Clarity: The proposed method is well motivated, the paper is carefully written, and clearly presented. I enjoyed reading the paper.

(2) Experiments: The experiments are also carefully designed. Both quantitative and human evaluation are provided. Quality examples are also shown.

Weaknesses:

(1) Novelty: The biggest concern that I have is its technical novelty. The proposed method is effective, but it acts more like a useful trick. Also, no theoretical justification is provided, but only some intuitions. So, I would say the novelty is indeed limited. However, given the comprehensive evaluation, and high writing quality, I lean to accept this paper due to its empirical contribution. It seems that this nucleus sampling method can be applied in a wide range of text generation applications.


** Minor **
Typo: In the line below Eqn. (2), "x \in V^{(k)}" => "x \in V^{(p)}", same typo in Eqn. (3).


**Experience Assessment:**

I have published one or two papers in this area.

**Review Assessment: Checking Correctness Of Derivations And Theory:**

I assessed the sensibility of the derivations and theory.

**Review Assessment: Checking Correctness Of Experiments:**

I assessed the sensibility of the experiments.

**Review Assessment: Thoroughness In Paper Reading:**

I read the paper at least twice and used my best judgement in assessing the paper.

---

> ### Author Response · Authors · 2019-11-14
> **Response (1/1)**
>
> Thank you for your positive assessment.
>
> -- Theoretical Grounding --
>
> While we don't have a theoretical proof of Nucleus Sampling, our paper does provide strong empirical evidence to justify truncated sampling in general and Nucleus Sampling in particular. Our most principled justification lies in analyzing the perplexity of generated text, which shows that, to match the perplexity of human written text, some form of truncated sampling has to be performed and that empirically this is correlated with generation quality. We suspect that this may be due to current large language models not fitting the underlying distribution optimally, but addressing that lies outside the scope of this paper.
>
>
> -- Novelty and Insight --
>
> In terms of novelty, we would like to highlight three main points. First, we provide insight into why truncation is necessary and how best to truncate the distribution of neural language models, analysis not performed in the papers that introduced Top-k sampling (only introduced last year) where it was described as a detail of decoding. Second, we provide the first side-by-side empirical analysis on how the quality of language generated by different LM decoding methods compares. Identifying the weaknesses and missing inductive biases of these methods will aid future work grappling with the theoretical implications of different methods. Finally, despite being "just another way of truncating the distribution", Nucleus Sampling provides a practical solution for generating high-quality text in various applications that mimics the human distribution more faithfully than competing methods.

---

### Official Review · AnonReviewer1 · 2019-10-23
**Official Blind Review #1**

**Rating:** 6

**Review:**

This paper is motivated by an observation that maximization-based decoding approaches such as beam search can lead to incoherent and repetitive sentences when open-ended long-form text generation based on neural language model such as GPT-2 is performed. To solve the problem, this paper proposes a sampling method called Nucleus Sampling. Similar to Top-k sampling, Nucleus Sampling truncates the probability distribution of the words in the vocabulary. Instead of re-normalizing the probabilities for the top-k words, Nucleus Sampling re-normalizes the original probabilities for the words with values above a pre-chosen threshold p. Some quantitative and qualitative results show that the proposed sampling method can generate long-form texts with some nice properties.

Pros:

The problem addressed in this paper is highly interesting, and the proposed method is simple and intuitive. The paper is well motivated and the method is clearly presented.

Extensive quantitative and qualitative experiments are conducted to compare different sampling methods.

Cons:

1) Although the raised problem in this paper is interesting, the proposed Nucleus Sampling seems to be a trivial variant of Top-k sampling. With a reasonably large k suitable for different practical problems in question, it is unclear that Nucleus Sampling produces significant advantages over commonly used Top-k sampling.

2) The argued difficulty in choosing k in Top-k sampling is not that different from that of choosing the threshold p in Nucleus Sampling.

3) In section 4.3, the argument that natural language rarely remains in a high-probability zone is questionable. This happens only because our current neural language models are not well-specified for generating long texts and modeling long-range contexts.

4) In section 6.2, the qualitative comparison between Nucleus Sampling and Top-k sampling might be caused by randomness. With a large k, there is no technical barrier that prevents Top-k sampling from generating the sentences produced by Nucleus Sampling.

5) A recent stochastic beam search method based on Gumbel-max-k (Kool, Hoof, and Welling, ICML 2019) should be discussed and compared.

In summary, although the studied problem in this paper is highly interesting, the proposed Nucleus Sampling is not technically significant compared to Top-k sampling.

**Experience Assessment:**

I have published one or two papers in this area.

**Review Assessment: Checking Correctness Of Derivations And Theory:**

I carefully checked the derivations and theory.

**Review Assessment: Checking Correctness Of Experiments:**

I carefully checked the experiments.

**Review Assessment: Thoroughness In Paper Reading:**

I read the paper thoroughly.

---

> ### Author Response · Authors · 2019-11-14
> **Response (1/2)**
>
> Thank you for your positive comments on our paper's motivation, presentation and evaluation. We gladly respond to your concerns:
>
> -- Novel Insights on Top-k Sampling and Beyond -- (Re: Con 1)
>
> We would like to emphasize that a primary contribution of this paper is the analysis into why truncation (such as Top-k sampling or Nucleus sampling) works and why it is necessary at all, analysis not performed in the papers that introduced Top-k sampling (introduced only as of last year), where it was described as a minor detail of decoding. We provide the first side-by-side empirical analysis and insight into the quality of language generated by current LM decoding methods to show that the tail of the distribution is unreliable. Identifying the weaknesses and missing inductive biases of these methods will aid future work grappling with the effects of different training and decoding methods.
>
> -- Comparison with Top-k Sampling -- (Re: Cons 1 & 2)
>
> When predicting the next word in a sequence, there will usually be a set of words that are plausible continuations, and a (usually much larger) set of words that are implausible (based on grammar or semantics). Nucleus Sampling captures the intuition that the size of the set of plausible next tokens will vary across different contexts, and can be approximated based on the probability distribution, rather than assuming a fixed-sized shortlist, as is the case with top-k sampling.
>
> Top-k sampling with a large value of k will cover most plausible next tokens, but also in some cases include inappropriate candidates that Nucleus Sampling would have excluded. Renormalization will increase those inappropriate candidates' probability of being sampled, which can degrade generation quality. This is further motivated in section 3.2 and Figure 4.
>
> Importantly, in this paper we provide extensive analysis showing why we need to perform truncated sampling to generate high-quality text from current large language models, which the papers proposing top-k sampling did not provide. Our analysis, for example, enables us to show quantitatively that top-k sampling works better with larger values of k than commonly used.
>
> -- Issues with Using Large k in Top-k Sampling -- (Re: Con 4)
>
> It is true that with a large enough k, Top-k can theoretically produce any sentence Nucleus Sampling can. In fact, pure sampling subsumes both Top-k and Nucleus Sampling in this sense. The problem, however, is that when using Top-k sampling with k large enough to generate sentences produced by Nucleus Sampling, it is also capable of generating other sentences with low coherence.
>
> -- Ease of Choosing Decoding Hyper-parameters -- (Re: Con 2)
>
> We believe that the choice of p is more intuitive than the choice of k because it more directly relates to the intuition that the sets of plausible and implausible candidate token can be captured by the head and the tail of the probability distribution. Equally importantly, in this paper we offer automatic metrics for choosing either p or k that were lacking from previous work. Qualitatively the exact choice of p between 0.9 and 0.99 appears to make relatively little difference in generation quality. For top-k sampling, coherence deteriorates when k is too large, and in practice (through small-scale expert evaluations) we found it hard to find a value of k that performs well on our automatic metrics while being as coherent as text generated by Nucleus sampling. To clarify this point, we will add an expert evaluation in the final version.
>
>
> -- Underlying Uncertainty in Natural Language  -- (Re: Con 3)
>
> We agree that there is still room for improvement in underlying language models for generating long-form text. However, there will always be uncertainty in what to say next, because there is real underlying uncertainty in language itself: it is extremely unlikely that language models can achieve perplexities in the neighborhood of 1.5, which is the perplexity we show greedy and beam search generations have.

---

> > ### Author Response · Authors · 2019-11-14
> > **Response (2/2)**
> >
> > -- Comparison to Stochastic Beam Search -- (Re: Con 5)
> >
> > Stochastic beam search (Gumble-top-k Beam Search) was proposed with a different motivation -- obtaining (pure) samples from the original distribution in parallel. The Gumble-top-k method was proposed to make beam search stochastic without truncating the distribution (as in top-k sampling or standard beam search). Our empirical findings, however, suggest that neural language models are unreliable estimators of the tail of the vocabulary distribution. Thus we intentionally truncate the search process to the head distribution and show that this produces higher quality generations that are closer to the human distribution of language. As shown in our experiments with pure sampling, sampling from the full distribution produces text that is more incoherent than decoding methods that use truncation.
> >
> >
> > ---
> >
> > In conclusion, the contribution of this paper lies as much in its technical analysis of the problem of text generation with large language models as in proposing a particular method that is robust and works well in practice.

---

> > > ### Comment · AnonReviewer1 · 2019-11-14
> > > **Thanks for the response**
> > >
> > >
> > > I have carefully read all the responses from the authors. Considering the merits of the empirical analyses performed, I have raised my score from Weak Reject to Weak Accept.
> > >
> > >
> > > About stochastic beam search:
> > >
> > > I agree that stochastic beam search also samples from the original distribution. However, due to the nature of the beam search, it will remove the issues of the incoherence caused by independently sampling each word from the original distribution. It  should avoid generating repetitive sentences. This baseline should be compared and discussed.

---

> > > > ### Author Response · Authors · 2019-11-15
> > > > **Follow-up and Initial Stochastic Beam Search Results**
> > > >
> > > > Thank you for your quick reply and engagement with our response.
> > > >
> > > > We appreciate your acknowledgement of the empirical analyses performed: Our perspective is that such analyses are vital to understanding the current landscape of generation, and that the analysis of methods, metrics, and models are key to studying text generation more rigorously.
> > > >
> > > > It is true that picking the highest scoring sample generated by stochastic beam search may help to alleviate the incoherence caused by sampling from the tail. We have run stochastic beam search with a beam size of 4 (we did not have enough compute to run larger beam sizes within the given time window) and found the numbers to be very close to pure sampling:
> > > >
> > > >     1)  The perplexity of the language model on text generated by stochastic beam search is 21.19, very close to pure sampling (22.73) and much higher than the perplexity of human text at 13.08.
> > > >     2)  Self-BLEU4 is 0.30 matching the human distribution, where pure sampling was slightly too diverse at 0.28.
> > > >     3)  The Zipf Coefficient is 0.92, slightly lower than human text (0.93) where pure sampling matched the human distribution.
> > > >     4)  As you suggested, repetition is lower using stochastic beam search, with only 0.06% of generations ending in a repetition loop. However, this actually underestimates repetition in naturally occurring human text at 0.18%.
> > > >     5)  HUSE requires human labels, which we could not obtain due to the limited time window, but which we will include in the final version.
> > > >
> > > > These initial numbers, especially the high perplexity, suggest that the issue of the incoherence of pure sampling generations is still present in stochastic beam search. In the final paper we will also include multiple beam sizes for a comprehensive comparison.

---

### Official Review · AnonReviewer3 · 2019-10-24
**Official Blind Review #3**

**Rating:** 6

**Review:**

In the domain of language models, the paper introduces a new heuristic sampling method called top-p sampling, or nucleus sampling (NS). It is a variant of top-k sampling where the smallest k is selected to ensure the combined likelihood is no less than p. The paper centers on claiming and showing that the generated samples are of higher quality and more diverse than common alternatives such as beam search, pure sampling, top-k sampling, and low-temperature sampling.

While overall I think the proposed method is sound as an alternative to other heuristics such as beam search, I have reservations on the presentation and arguments made in the paper.

Pros:
1. NS is sound as a heuristic sampling method.
2. The paper contains many interesting experimental observations and I speculate that some of them will find future uses. For example, the selection of parameter values (not just for NS, also for top-k) and the nontrivial perplexity of generated text.

Cons:
1. The ultimate performance measure (open-ended generation) of “high quality” and “diversity” is very vague. It seems that the authors end up doing is to evaluate by high self-BLEU, HUSE, few repetitions, and perplexity. Furthermore, it is unclear why one _should_ train with cross-entropy (trying to match the distributions) and then rely on the sampling procedure to fulfill these desiderata (See also Min1 and Min2).
2. The comparison with beam search (BS) is not well motivated. BS is devised to find the maximal sentence and it is not stochastic. It seems out of place in the context of generating a “diverse” set of samples.
3. The arguments in the comparison with pure sampling is vague and sometimes misplaced. The key argument seems to hinge on the idea that the low likelihood tail is of “low confidence.” But this claim is problematic. If the estimate is wrong on the low probability tail, then so is the estimate on p(head) = 1-p(tail) by virtue of p being a probability measure.
4. The arguments in the comparison with top-k is vague and sometimes misplaced. The main argument against top-k is the “[d]ifficulty in choosing a suitable value of k” but the same can be said for choosing p. After all, top-k and top-p (NS) can be thought of as a variant of each other (by dynamically choosing k or p respectively). Moreover, in Figure 5, a selection for k value is suggested. I agree that this value might not _appear_ as intuitive as p, and maybe other works have chosen a smaller k than they should have, but similarly, people might intuitively choose too high a value for p (Figure 5).

Possible mistakes/typos:
1. (2), “>=“ -> ≥.
2. Figure 7, the human self-BLEU4 < human self-BLEU5 and that seems wrong, especially when all other bars show the opposite ordering.
3. In References, “Angela Fan, Mike Lewis, and Yann Dauphin. Hierarchical neural story generation. In ACL, 2018a” is duplicated.
4. In References, the citation of “Unifying human and statistical evaluation for natural language generation” is from NAACL 2019, not “2018.”
5. In References, the first names are shown as initials in “Sparse forward-backward using minimum divergence beams for fast training of conditional random fields.”

Questions:
1. In Table 1, how is Human perplexity estimated?

Minor issues:
1. Partly due to what the authors position NS to solve, i.e. open-ended generation, the core arguments is not as precise or rigorous as it could have been in my opinion. I feel that focusing on comparing NS to other heuristics as a heuristic might make the text appeal to a wider audience and the discussion more precise.
2. The distinction drawn between open-ended generation and directed generation is unpersuasive to me. In the context of language modeling, the former is to approximate a distribution (over an extended alphabet) whereas the latter is to approximate a conditional distribution (given the input). However, the most common formulation to solve the former is to decompose the distribution into a product of conditional distributions (1).
3. The caption in Figure 1 draws a misleading comparison. The “admirable” generation (presumably referring to the OpenAI blog post) was from the full GPT-2 model, not the initially released GPT-2-117M.

Please point out my misunderstanding directly. I am open to acknowledging them and revising my assessment.

**Experience Assessment:**

I have published one or two papers in this area.

**Review Assessment: Checking Correctness Of Derivations And Theory:**

I carefully checked the derivations and theory.

**Review Assessment: Checking Correctness Of Experiments:**

I assessed the sensibility of the experiments.

**Review Assessment: Thoroughness In Paper Reading:**

I read the paper thoroughly.

---

> ### Author Response · Authors · 2019-11-14
> **Response (1/2)**
>
> Thank you for your positive overall assessment. We gladly respond to your concerns and provide clarifications that we hope will clear up some potential misunderstandings:
>
>
> -- Evaluating Open-Ended Generation -- (Re: Con 1)
>
> Evaluating open-ended generation is a hard problem for which there are currently only partial solutions, but we think this should encourage rather than discourage further work towards proper evaluation. Developing better models and better evaluation criteria go hand in hand—while we propose several criteria in the paper, we do not believe that any one of them is sufficient to use directly as a training criteria.
>
>
> -- Why Use Cross-Entropy Loss? -- (Re: Con 1)
>
> Large language models such as GPT-2 are the best currently available models for general purpose text generation. While it is possible that training criteria other than cross-entropy could result in a better model, most other currently available criteria are not differentiable and not as scalable. In practice other training criteria such as GANs have been shown to lead to worse generation quality than cross-entropy training (see references in 2.1).
>
>
> -- Why Sampling is Necessary for Good Generation -- (Re: Con 1)
>
> To generate text from large language models we believe that some form of sampling is required precisely because in open-ended generation maximum probability texts do not match the human distribution of text. Indeed, we show that to match the human distribution (in terms of perplexity) it is actually better to perform truncated sampling than pure sampling (at least with current models).
>
>
> -- Why Compare to Beam Search? -- (Re: Con 2)
>
> The reason for comparing to beam search is that it has indeed been used in recent conditional open-ended generation work (Ammanabrolu et al., 2019, Anonymous, 2019). Furthermore, we are interested in finding the best method to generate high quality text with the same diversity of vocabulary as human text, rather than generating a diverse set of samples. Beam search could reasonably be a way to achieve that, although in practice we show that the quality of text that it generates is deficient.
>
>
> -- Why is the Tail of the Distribution Considered Unreliable? -- (Re: Con 3)
>
> Our hypothesis is that the (relative) probability estimates of words within the tail are inaccurate (either too high or too low), rather than the overall p(tail). Pure sampling generates text which does not match the human distribution (as measured by perplexity) because when semantically inappropriate words (whose probability estimates are presumably too high) are sampled from the tail, that throws the sampled sequence off the correct distribution, leading to incoherence in practice. Therefore, lacking a better underlying model, the best solution is not to sample from the tail.
>
>
> -- Novelty of Analysis -- (Re: Con 4)
>
> Firstly, in this paper we offer automatic metrics for choosing either p or k — analysis lacking from previous work — which enables us to show that higher values of k should be used. We think, conceptually, that having a dynamic k fixed by p is better than having a dynamic p fixed by k (as explained in section 3.2 and figure 4). Qualitatively the exact choice of p between 0.9 and 0.99 appears to make relatively little difference in generation quality. For top-k sampling, coherence deteriorates when k is too large, and in practice (through small-scale expert evaluations) we found it hard to find a value of k that performs well on our automatic metrics while being as coherent as text generated by Nucleus sampling. To clarify this point, we will add an expert evaluation in the final version.
>
> -- Perplexities of Human Text -- (Re: Question 1)
>
> We report the perplexities of the original model on text produced by each method. The column with "human" perplexity is the perplexity of the original model on the human-written continuations in our experimental setup.
>
>
> -- Scope of the paper -- (Re: Minor Issue 1)
>
> To clarify: we don't wish to claim that Nucleus Sampling is something other than a heuristic or that it "solves" open-ended generation. We do aim to give a better understanding of the problem of open-ended generation, the various methods that have been proposed to address it, and ways to evaluate them, rather than just comparing Nucleus Sampling to other generation strategies.
>
> -- Open-ended vs directed generation -- (Re: Minor Issue 2)
>
> The distinction between open-ended and directed generation is not the same as the distinction between conditional  and unconditional generation—indeed we use a conditional setting for open-ended generation in this paper (section 4.1). The distinction is that in open-ended generation there is much more uncertainty in the conditional distributions, which means that in practice decoding methods that work for directed generation, where the output is close to a direct transformation of the input (and therefore has low uncertainty), do not work for open-ended generation.

---

> > ### Author Response · Authors · 2019-11-14
> > **Response (2/2)**
> >
> > -- Comparing different versions of GPT-2 -- (Re: Minor Issue 3)
> >
> > Radford et al. also released high-quality generations from the smaller GPT-2 model - our focus is on the decoding strategy used rather than the model choice. We have updated Figure 1’s caption to make this clear. The full GPT-2 model was not publically available at the time of submission.
> >
> > References
> >
> > Ammanabrolu et al., 2019. "Guided Neural Language Generation for Automated Storytelling." Proceedings of the Second Workshop on Storytelling.
> >
> > Anonymous, 2019. "Neural text generation with unlikelihood training." https://openreview.net/forum?id=SJeYe0NtvH

---

### Decision · Program_Chairs · 2019-12-19

**Decision:**

Accept (Poster)

**Comment:**

This paper presents nucleus sampling, a sampling method that truncates the tail of a probability distribution and samples from a dynamic nucleus containing the majority of the probability mass. Likelihood and human evaluations show that the proposed method is a better alternative to a standard sampling method and top-k sampling.

This is a well-written paper and I think the proposed sampling method will be useful in language modeling. All reviewers agree that the paper addresses an important problem.

Two reviewers have concerns regarding the technical contribution of the paper (i.e., nucleus sampling is a straightforward extension of top-k sampling), and whether it is enough for publications at a venue such as ICLR. R2 suggests to have a better theoretical framework for nucleus sampling. I think these are valid concerns. However, given the potential widespread application of the proposed method and the strong empirical results, I recommend to accept the paper.

Also, a minor comment, I think there is something wrong with your style file (e.g., the bottom margin appears too large compared to other submissions).